# Intelligent Feature Selection for ECG-Based Personal Authentication Using Deep Reinforcement Learning

**DOI:** 10.3390/s23031230

**Published:** 2023-01-20

**Authors:** Suwhan Baek, Juhyeong Kim, Hyunsoo Yu, Geunbo Yang, Illsoo Sohn, Youngho Cho, Cheolsoo Park

**Affiliations:** 1Department of Computer Engineering, Kwangwoon University, Seoul 01897, Republic of Korea; 2Department of Computer Science and Engineering, Seoul National University of Science and Technology, Seoul 01811, Republic of Korea; 3Department of Electrical and Communication Engineering, Daelim University, Kyoung 13916, Republic of Korea

**Keywords:** ECG, authentication, biometrics, reinforcement learning, feature selection, hyperparameter optimization

## Abstract

In this study, the optimal features of electrocardiogram (ECG) signals were investigated for the implementation of a personal authentication system using a reinforcement learning (RL) algorithm. ECG signals were recorded from 11 subjects for 6 days. Consecutive 5-day datasets (from the 1st to the 5th day) were trained, and the 6th dataset was tested. To search for the optimal features of ECG for the authentication problem, RL was utilized as an optimizer, and its internal model was designed based on deep learning structures. In addition, the deep learning architecture in RL was automatically constructed based on an optimization approach called Bayesian optimization hyperband. The experimental results demonstrate that the feature selection process is essential to improve the authentication performance with fewer features to implement an efficient system in terms of computation power and energy consumption for a wearable device intended to be used as an authentication system. Support vector machines in conjunction with the optimized RL algorithm yielded accuracy outcomes using fewer features that were approximately 5%, 3.6%, and 2.6% higher than those associated with information gain (IG), ReliefF, and pure reinforcement learning structures, respectively. Additionally, the optimized RL yielded mostly lower equal error rate (EER) values than the other feature selection algorithms, with fewer selected features.

## 1. Introduction

Security issues have been considered as a critical factor for the Internet of Things (IoT) owing to privacy challenge concerns [1,2,3,4]. For instance, a smart health card generated based on an IoT platform may enhance patient security and privacy information. However, when it is hacked, security issues are raised, such as theft risk, loss, insider misuse, and unintended behavior. Knowledge-based authentication methods rely on users’ memories, whereas token-based authentication methods utilize an external device [2,5].

For example, knowledge-based authentication methods use a personal identification number (PIN) and an identity (ID)/password, and token-based ones provide one-time passwords (OTPs) and short message services (SMSs) to the users. However, both approaches could be vulnerable to a brute-force dictionary attack, that is they can be guessed, duplicated, lost, or stolen. In particular, knowledge-based authentication methods could be attacked by hackers who may guess the users’ family name, birthday, or anniversary, and token-based methods could be critically risky when the external device is lost or stolen [6,7].

To solve these issues, researchers are investigating different personal authentication approaches using biometric data. With a biometric authentication system, users do not have to remember complex passwords or hold tokens, but may access the system using unique features of their own bodies that would be difficult to be cloned, lost, or stolen [6,7,8,9,10,11,12,13]. However, some types of biometrics, such as fingerprints, irises, and faces, are still vulnerable to attack. Fingerprints could be imitated and duplicated with silicone [14,15]; the iris features could be reproduced with contact lenses and printing [16]; the face could be easily fabricated with a photograph [17]. In addition, these biometric features have a critical flaw in that they cannot be remedied if they are damaged [6,13].

The electrocardiogram (ECG) is used as one of the biometrics. It is an electrical signal generated by the sinoatrial node in the heart to stimulate the cardiac muscle to contract and relax. It consists of various peaks referred to as P, Q, R, S, and T waves (see Figure 1). Compared with other biometrics, the ECG signals cannot be easily reproduced and have higher reliability, entropy, and randomness [18,19,20,21,22]. Additionally, ECG signals are affected by various other factors, including age, gender, physical condition, structure, and obesity [23,24,25,26]. To extract ECG signal features for the implementation of the authentication system, data-driven convolutional neural network (CNN) models have either been designed [27,28,29] or feature-engineering approaches have been applied based on predefined fixed models [30,31,32,33].

The extracted features from a single-lead ECG signal have been proven to provide reliable authentication results [34,35,36]. It has also been reported that the long-term stability of the features is guaranteed for several days or even years [34,37,38]. This study also explored the long-term stability of the ECG features for a personal authentication system using ECG signals that were recorded for six days. Additionally, this study identified the ideal ECG features that were considered the most significant for the classification of a user among others. It was found that the biometric authentication task that uses a high number of significant features, also known as “costly features”, performed better than the one that used all the features extracted from the biometric signals without taking into consideration their significance in relation to the task [39,40,41,42].

However, authentication with these ECGs has some limitations. Violent activities such as exercise may change the ECG features [43]; drugs such as caffeine may change the ECG features [19]; emotional changes may cause difficulties in ECG-based authentication [44]; the heart rate may change every day [45]. In this paper, experimental data were created to design robust models for problems caused by ECG that vary daily among these challenges. Unlike conventional data, the data used in this paper comprise different cardiac data over a continuous six-day period of one subject. The model optimized through the data will have the strength of having relatively robust results for daily varying ECG signals.

Among the algorithms used to search the costly features, we mainly applied the reinforcement learning (RL) algorithm [46] to ECG-based personal authentication. Recently, RL has achieved considerable performance improvements with the help of deep learning models, yielding state-of-the-art results in various areas, such as healthcare, autonomous driving, and resource management [47,48,49,50,51,52]. In addition, many studies have been conducted for feature selection using RL owing to its promising performance for the optimization [53,54,55]. The deep neural networks in RL, commonly referred to as deep Q-learning [56], were manually constructed in previous studies [48,51,57]. However, the performance of deep neural networks will vary depending on their architectures; additionally, they were developed mostly based on the developer’s experience and intuitions and may have suboptimal architectures. Thus, the networks in RL are automatically optimized using the Bayesian optimization hyperband (BOHB) method [58]. In this study, BOHB optimized the layers of the neural networks, the number of nodes in each layer, the learning rate, and the optimizer in the RL algorithm. As a benchmark test, the costly conventional feature selection algorithms, namely ReliefF [59,60] and information gain (IG) [61], were compared. The former is a Manhattan-distance-based feature selection algorithm that selects the significant features by calculating the sum of the distances among the instances of the features. The latter is an entropy-based feature selection algorithm computing the entropy of each feature and determines the significant features based on the calculated entropy.

This study is structured as follows. Section 2 elaborates on the RL deep Q-network (DQN) and BOHB algorithms for optimal feature selection. Section 3 describes the experimental methods, preprocessing, and feature extraction. The conducted experiments to demonstrate the effectiveness of the optimization of DQN as a model-independent classifier via BOHB are described in Section 3.3. Some experiments using different models are described in Section 3.4 for comparison with the optimized RL model and other costly feature selection algorithms. In Section 4, the authentication results (using RL and BOHB) are provided based on the benchmark tests with the conventional methods for both experiments.

## 2. Materials and Methods

### 2.1. Costly Features in IoT Environment

In an IoT environment, limited resources such as memory, computation, and power have always been issues [62,63,64,65,66,67]. In particular, the classification problem for a personal authentication system pertaining to wearable devices is also limited by these issues; this is referred to as classification with costly features (CwCF). Previously, the RL algorithm was designed to solve the CwCF issue to minimize the expected classification errors with incurred costs [46].

### 2.2. Deep Q-Network

RL is a machine learning approach in which an agent finds the optimal action and policy based on rewards from the environment. RL consists of Markov decision processes (MDPs) [68]. The elements of an MDP have a state *s*, action *a*, reward *r*, and depreciation rate of γ. Specifically, *s* denotes the current state, and *a* is the action taken in *s*. In turn, *r* is the reward obtained from the environment when the agent takes an action, and γ is the reliability in future rewards whose values range between 0 and 1.

Q-learning tries to identify the optimal policy of the MDP by updating the Q-function [69]. At the beginning of each episode, an agent moves from the current state *s* to the direction defined by the current action, *a*. The agent receives a reward *r* from the environment, yields a Q-value for the next action $a^{\prime }$, and obtains the maximum Q-value from the next state $s^{\prime }$. Subsequently, the Q-value is updated by multiplying the maximum Q-value and learning rate α according to Equation (Equation 1).
(1)$$Q^{update}\left(s_{t},a_{t}\right)⇐\left(1-\alpha \right)Q\left(s_{t},a_{t}\right)+\alpha \left(r_{t+1}+\gamma max_{a^{\prime }}Q\left(s^{\prime },a^{\prime }\right)\right)$$

The rewards are obtained at t+1 based on the current state and environment.

The DQN applies deep neural networks to Q-learning to approximate the Q-value in more complicated environments than that of conventional Q-learning [56,70]. The loss function of the model calculates the mean-squared error (MSE) L(θ) based on Equation (Equation 2):(2)$$L\left(\theta \right)={\left(r_{t+1}+\gamma max_{a^{\prime }}Q\left(s^{\prime },a^{\prime };\theta^{-}\right)-Q\left(s_{t},a_{t};\theta \right)\right)}^{2}$$
where $\theta^{-}$ are the parameters of the target network, which are fixed. The target network is updated in every predefined number of epochs. In addition, the DQN utilizes the experience replay method, wherein samples, including the set ($s_{t}$, $a_{t}$, $r_{t+1}$, $s_{t+1}$, $a_{t+1}$), are stored in memory and a specific number of samples are randomly chosen to train the networks. This could solve the issue of dependence on consecutive samples or avoid unnecessary feedback loops.

### 2.3. Costly Feature Selection Using RL

The costly feature selection is described as follows. The variable (x,y)∈D denotes one of the samples from the data distribution *D*, where the vector *x* contains *n* input features, $f_{i}\in F=f_{1},\ldots ,f_{n}$, and *y* is its class label. In one episode, the environment randomly selects one data sample from *D*, and the agent sequentially selects the features and classes with the highest Q-value [46]. The environment is represented by a partially observable MDP (POMDP) [71], which, unlike an MDP, provides the agent with limited information about the environment. State $s=(x,y,\bar{F})\in S$ is denoted by a sample (x,y), the state space S, and the agent-selected features $\bar{F}$. In action $a\in A(A=A_{c}⋃A_{f})$, $A_{f}$ is an action taken to conduct the classification, and $A_{f}$ is an action taken to select a feature in a feature set. The episode ends when the agent selects a classification action, $A_{c}$, and receives a reward of 0 if it is correctly classified and −1 if it is incorrect. When the agent selects an action $A_{f}$, to select a feature, it receives a reward of $-\lambda c(f_{i})$, where $c(f_{i})$ is the cost for $f_{i}$. The reward function r:S×A→R is in accordance with S and A and is represented mathematically as follows.
(3)$$r\left(\left(x,y,\bar{F}\right),a\right)=\left\{\begin{array}{cc} -\lambda c(f_{i}) & \mathrm{if}\;a\in A_{f},a=f_{i}\\ 0 & \mathrm{if}\;a\in A_{c}\;\mathrm{and}\;a=y\\ -1 & \mathrm{if}\;a\in A_{c}\;\mathrm{and}\;a\neq y \end{array}\right.$$

The value of λ provides a trade-off between the precision and average cost for this RL model. As λ increases, the cost is reduced and the focused episode becomes shorter. The transition function is defined as t:S×A→S⋃T.
(4)$$t\left(\left(x,y,\bar{F}\right),a\right)=\left\{\begin{array}{cc} T & \mathrm{if}\;a\in A_{c}\\ \left(x,y,\bar{F}⋃a\right) & \mathrm{if}\;a\in A_{f} \end{array}\right.$$
where T is the terminal state. When the agent selects a feature as an action, it adds the currently selected feature to $\bar{F}$. If the agent selects an action to derive the classification result, it ends the episode.

In this paper, we designed a feature selection model using the DQN algorithm one of the promising RL models. If only the feature is placed in the action of the DQN model, the model acts as an optimizer [72], but by giving both feature and subject number, the model could possibly perform both feature selector and classifier functions as a pure RL model [46]. The procedure of this algorithm is shown in Algorithm 1.

**Algorithm 1** Procedure of DQN Optimizer and Classifier.  1 : Initialize replay memory   2 : Initialize action value function Q with random weights  3 : for ϵ = 1, M do  4 :     for t = 1, T do  5 :           With probability epsilon, select a random action   6 :           if random action is feature:  7 :                   Execute action in emulator, and observe reward   8 :                   Set state and preprocess policy  9 :                   Store transition in replay memory 10 :                  Perform a gradient descent step 11 :           if random action is subject number:  12 :                   Execute action in the emulator, and observe reward      13 :                   Set state and preprocess policy 14 :                   Store transition in replay memory 15 :     end for 16 : end for 

### 2.4. Hyperparameter Optimization

The performance of machine learning algorithms relies on internal hyperparametric settings. A machine learning algorithm could be represented as a function g:X→R and its hyperparameters x∈X. The hyperparameter optimization (HPO) task aims to identify the optimal hyperparameters $x^{*}\in {\mathrm{argmin}}_{x\in X}g\left(x\right)$. However, most machine learning algorithms cannot observe g(x) owing to its randomness and uncertainty and, thus, assume that it is observable only based on noisy observations y(x)=g(x)+ϵ, with $\epsilon ∼N(0,\sigma_{noise}^{2})$ [58,73,74].

### 2.5. Bayesian Optimization

In each iteration *i*, Bayesian optimization (BO) builds a probability function p(g|D) to model the objective function *g* using the Gaussian process, which is based on the already known (observation) dataset $D=\{\left(x_{0},y_{0}\right),\ldots ,\left(x_{i-1},y_{i-1}\right\})$ [58,73,75]. BO applies the acquisition function a:X→R based on the current model p(g|D), and the model considers a tradeoff between the processes of exploration and exploitation; iterations are conducted based on the following three steps:

(1) Select an observation at which the acquisition function is maximum $x_{select}={\mathrm{argmax}}_{x\in X}a\left(x\right)$;

(2) Evaluate the objective function $y_{select}=g\left(x_{select}\right)+\epsilon$;

(3) Augment the dataset with the selected observation, $D=D\cup (x_{select},y_{select})$.

During the process, the model tries to identify the best observation $x_{best}=argmin_{x\in D}g\left(x\right)$.

### 2.6. Hyperband

Hyperband is a resource allocation problem-solving method executed in a purely exploration adaptive manner and constitutes a configuration evaluation approach based on the formulation of the hyperparameter optimization [58,76]. This method uses a principled early stopping strategy to allocate resources; the strategy aims to quickly identify superior hyperparameters by examining larger-scale hyperparameter configurations instead of using a strategy based on the uniform training of all configurations.

### 2.7. BOHB Hyperparameter Optimization

BOHB [58] is an HPO method that combines BO and the hyperband (HB). The BO process in BOHB uses a tree Parzen estimator (TPE) [77], which models a density function using a kernel density estimator. Algorithm 2 displays the procedure of the BOHB algorithm. Both feature selection using the BO algorithm and hyperparameter optimization using the HB algorithm are conducted simultaneously. Although the algorithm follows the budget selection approach of the HB, it guides the search by replacing a random sampling using a BO component. BOHB often searches for a good solution at a much faster rate than BO and converges to the best solution at a much faster rate than hyperband. In this study, the hyperparametric optimization method was applied to determine the number of hidden layers, learning rate, and optimizer for the DQN. The entire procedure of this algorithm is shown in Figure 2. State *s* consists of tuples $\left(\bar{x},m\right)$, where $\bar{x}$ is the masked vector of the original η, and is defined by the mask vector *m*; the latter is composed of (0, 1) and is responsible for the index of the selected feature.

**Algorithm 2** Procedure of BOHB algorithm.  1 : Input the number of maximum budget R, setting η  2 : Initialization the number of setting S${}_{m}ax$ = ceil(logη R)  3 : for s = S${}_{m}ax$ to 0 do  4 :     set current Configuration A  5 :           for i = 0 to s do  6 :                  Select hyperparameter Configuration A${}_{i}$     7 :                  Get loss L using Configuration A${}_{i}$  8 :                  A = min(L(A), L(A${}_{i}$))  9 :           for t = 1 to T
do 10 :                  Calculate a probability function p(g|D) using Gaussian process 11 :                  Select observation where x${}_{s}elect$ = $argmax_{x}\in$*x* a(x) 12 :                  Evaluate the objective function y${}_{s}elect$ = g(x${}_{s}elect$) + ϵ 13 :                  Add dataset D = D ∪ ( x${}_{s}elect$, y${}_{s}elect$ )  14 :                  Update best observation x${}_{b}est$ = argmin${}_{x}\in \;Dg\left(x\right)$ 15 :     end for 16 : end for 


(5)
$${\bar{x}}_{i}=\left\{\begin{array}{cc} x_{i} & \mathrm{if}\;f_{i}\in \bar{F}\\ 0 & \mathrm{otherwise} \end{array}\right.$$



(6)
$${\bar{m}}_{i}=\left\{\begin{array}{cc} 1 & \mathrm{if}\;f_{i}\in \bar{F}\\ 0 & \mathrm{otherwise} \end{array}\right.$$


The agent selects $Q_{class}$ or $Q_{feature}$ corresponding to the current state. The previously selected features cannot be chosen again owing to the mask vector *m*.

## 3. Experiments

### 3.1. ECG Measurement Experiments

An experiment was conducted to generate a dataset to train and evaluate the proposed model. To record the ECG signals from the subjects, a commercially available real-time recording system was used (MP36, Biopac Systems, Goleta, CA, USA) at a sampling rate of 1000 Hz. Eleven subjects were invited and their ECG signals were recorded for 10 min for six days at random times from 10:00 a.m. to 4:00 p.m. The subjects were seated in a comfortable chair in an enclosed space and kept in a relaxed state. During the experiment, ECG signals were recorded from the left wrist with reference to the right wrist and with a ground electrode on the ankle, a configuration known as the driven-right leg [30,78]. A bandpass software filter with a finite impulse response (FIR) filter between 1 Hz and 35 Hz was used to minimize the ambient noise components [79,80,81]. Figure 3 illustrates the noise reduction using the bandpass filter process. The subjects had an average weight of 73.20 kg (±9.2 kg), an average height of 174.6 cm (±6.8 cm), an average BMI of 23.93 (±1.93), and a average age of 27.4 (±5.1). A group of 11 male subjects participated in the experiments.

### 3.2. Feature Extraction

The features of the ECG signals were extracted using the information of the P, Q, R, S, and T peaks. For the automatic peak extraction, the Pan and Tompkins [82] algorithm was applied to the ECG signals. They were defined based on the amplitudes, intervals, slopes, and angles of the peaks; in total, 31 features were derived in combinations with all peak points [30,31,32,33]. Figure 1 displays a typical ECG pattern with the five peaks; the extracted features are listed in Table 1. The features of the amplitude were extracted by the ratios among the peaks.

### 3.3. Evaluation of BOHB-Optimized DQN Authentication Algorithm

Experiments were conducted to investigate whether the BOHB optimization of the DQN could improve the authentication performance. These experiments evaluated the independent performance of the DQN model with BOHB used as an RL classifier. BOHB was applied to optimize the DQN for the RL-based, costly feature selection algorithm. In this experiment, the hyperparameters in the DQN (to be optimized) included the number of layers, nodes in each layer, learning rate, optimizer, and stochastic gradient descent (SGD) momentum, as summarized in Table 2. The minimum budget of BOHB was set to one and the maximum budget to nine. Table 3 shows the number of beat data generated by each subject for training and evaluating the proposed model. During the training process, the synthetic minority oversampling technique (SMOTE) [83], which is an oversampling method for the data augmentation, was applied to improve performance during the training process [45]. It is the method of generating a new sample using the distance between selected samples within the same group by applying the K-nearest neighbor (KNN) [84] algorithm. The dataset recorded from the 1st to 5th days was trained, and the 6th day of recordings were tested. Additionally, five-fold cross-validation was used to evaluate the generalization of the trained DQN model.

### 3.4. Evaluation of Costly Feature Selection Algorithms

This experiment was designed to evaluate the costly feature selection performance of the RL model. In this experiment, the proposed RL-based, costly feature selection algorithm was compared with the conventional feature selection methods, including ReliefF [59,60] and IG [61]. Therefore, the RL algorithm model (DQN) was only utilized for the selection of costly features; the selected features were then fed into the conventional classifiers for evaluation. Furthermore, an effective classifier for the authentication problem was also evaluated using support vector machines (SVMs) [85] and random forest (RF) [86]. The SVM and RF were chosen based on their promising performances in various machine learning problems, such as featured-based classification [87,88], image classification [89,90], and anomaly detection [91,92]. To evaluate the SVM and RF machine learning algorithms, personal authentication for input on the 6th day was conducted based on the sequentially cumulative trained model from 1 to 5 days of 11 subjects’ data. It was utilized as a training and verification dataset from the 1st to 5th days of subject data, and the model’s performance was tested with data from 6th day of ECG signals.

## 4. Results

### 4.1. Results of BOHB Optimized DQN Authentication Algorithm

Figure 4 shows the testing accuracy of the ECG-based authentication task using the optimized and non-optimized DQN models. The feature selection and authentication were performed simultaneously through the DQN models, and each training dataset was incrementally increased with the following day’s dataset. The results are the averaged authentication results across all subjects. Overall, the average accuracy of the non-optimized RL method was 95.3%, while that of the optimized method was 97.4%. In particular, note the significant improvement of the F1-score using the optimized DQN (95.2%), which was 10.9% higher than that of the non-optimized DQN (84.2%).

### 4.2. Results of Costly Feature Selection Algorithms

Figure 5 and Figure 6 depict the classification results including the performance indices, accuracy, and F1-scores, using the SVM and RF, respectively, where the indices were the averages of 10 simulation repetitions for the test dataset. In the figures, four different costly feature selection algorithms are compared: DQN with BOHB optimization (optimized RL), DQN without any optimization process (RL), and the ReliefF and IG feature selection algorithms. For the optimized DQN classifier, the model with the highest validation accuracy was chosen, and the feature selection was conducted. The x-axis of the subplots in Figure 5 and Figure 6 displays the average numbers of the selected features for the model’s final decision, while the y-axis displays the accuracy and F1-score performance indices. In Figure 5, the use of the SVM classifier of the optimized DQN algorithm outperformed the other feature selection algorithms with accuracies of 96.5%, 97.2%, 98.1%, 98.3%, and 98.5%, and F1-scores of 75%, 75.9%, 74.8%, 84.6%, and 91.1%. The numbers of the selected costly features were approximately equal to 3.9, 3.7, 2.9, 4.5, and 5.2.

In Figure 6, the accuracy and F1-score are shown to be higher using the 1st–3rd day training dataset based on the use of the RF as the classifier and the optimized RF as the feature selection algorithm, despite the fact that the accuracy and F1-score of ReliefF (using the 1st–5th days training dataset) were higher than those of the optimized DQN model and that they required more features, that is 4.3, 5.2, and 6.2 in the case of the optimized DQN and 6.5, 8.2, and 7.8 in the case of ReliefF.

As shown in Figure 5 and Figure 6 the “Optimized Reinforcement Learning” method proposed in this paper reported higher accuracy and F1-score when using the same number of features compared to other methods. These were the result of the model’s selection of the most-optimized features from possible combinations of ECG features, demonstrating that the model’s optimization through reinforcement learning was effective to improve the authentication task.

The equal error rate (EER) was determined by the false acceptance rate (FAR and false rejection rate (FRR) when they are equal [93]. The FAR and FRR are calculated using Equation (Equation 7), where FP, TN, FN, and TP denote false positive, true negative, false negative, and true positive outcomes, respectively.
(7)$$FAR=\frac{FP}{FP+TN},\;FRR=\frac{FN}{FN+TP}$$

Figure 7 illustrates the EER results of all combinations among the costly feature selection algorithms and classifiers. The x-axis of the subplot in Figure 7 displays the average number of selected features for the model’s final decision, while the y-axis displays the EER value. The best performance with the fewest number of features and lowest EER could be obtained using the optimized DQN with SVM using the training dataset recorded from the 1st to the 3rd day, that is using approximately three features and an EER of 4.7%. Although the lowest EER was obtained with ReliefF and RF using all five-day training datasets, more features (1.5-times) were used than those of the second-best method (see Figure 7e).

## 5. Discussion

In this study, a personal authentication task was conducted based on ECG signals recorded for 6 days. The ReliefF and information gain algorithms are representative conventional feature selection methods, which are simpler than the optimized and pure RL methods. Although the accuracy incrementally improved as shown in Figure 5 and Figure 6, they are not reliable in terms of the F1-score and equal error rate performances. The optimized model based on the method proposed in this study yielded high performance compared with the other conventional feature selection approaches. This demonstrates that the ECG signal could be feasible in implementing the biometric authentication system in our daily lives.

The optimized RL using BOHB produced the most-efficient and best performance in selecting costly features compared with other conventional methods, as proven by the accuracy, F1-score, and EER outcomes of the authentication tasks. This also proved the effectiveness of the model optimization process, commonly referred to as the automatic machine learning (Auto ML) process [94], based on the feature selection tasks using the RL algorithms. The results in Figure 5 and Figure 6 show that the proposed costly feature selection method could yield different performances depending on the classifier of the machine learning algorithm. The proposed approach was clearly improved by the SVM model compared with the RF model. This implies the optimal combination of the costly feature selection method and the classifier for the ECG-based authentication task. The RL model performed best with the two machine learning classifiers, thus implying that the costly feature selection method proposed in this study could be optimal in yielding the improvement of the authentication performance. This could be supported by the various optimization studies based on the RL algorithms, which typically perform better than other approaches [8,48,95].

It is noted that the suggested feature selection method outperformed the others (see Figure 5 and Figure 6). In particular, from the perspective of the F1-score, the non-optimized RL model yielded similar results to the other traditional feature selection methods, while the optimized BOHB-based model yielded improved results. This trend may indicate that the optimized DQN model could select significant features even with a small number of datasets. In addition, the proposed model selected a relatively small number of features compared with those selected by the other methods. During the learning process of the RL algorithm, the received rewards decreased as the number of learning episodes increased; this resulted in an automatic termination of the feature selection process at the appropriate level of training, while the traditional models require stopping the selection of features manually based on the experience of the model designer. This automatic stopping property of the RL algorithm could provide an efficient approach to saving learning and resources.

Figure 8 displays the number of subjects who selected the features through the optimized RL by increasing the training dataset. Note that some specific features, such as the “QS slope”, include more subjects than the others as the training data increase. The model selected the QS slope, F1-score, and EER using the training dataset recorded from the 1st to 5th days and produced the best results with the highest accuracy. Additionally, the selection of the QR slope, RT amplitude, and PT amplitude gradually increased as more training datasets were included.

We recorded ECG data from the subjects for six days. Among them, the data from the 6th day were used as the test dataset and were evaluated based on various feature selection and classification algorithms. Among the various optimal feature selection algorithms, the BOHB-optimized DQN algorithm produced the most-improved results compared with the SVM model. When there were adequate data for the training, the accuracy converged to values greater than 90%. The results produced by the optimal number of features could suggest the implementation of the ECG-based personal authentication model with a tight-sized structure in edge devices, such as smartwatches and mobile devices. To demonstrate this implementation, the machine-learning-based algorithm (SVM, RF) algorithm proposed in this paper was run on the Raspberry Pi4 board, confirming that it can be processed in less than 10 seconds per about 1000 heartbeats. The reason this can be implemented is that we optimized the costly features and classified them using relatively light-sized classifiers, rather than optimizing complex neural networks. Thus, this study demonstrated that this personal authentication model could be utilized in various embedded equipment types or low-power environments.

## 6. Conclusions

In this study, an RL-based personal authentication model and its optimization were proposed. These yielded significant performance enhancements compared with the conventional methods. Furthermore, they can be applied to various embedded systems with machine learning classifiers with relatively low resource consumption, such as the SVM and RF algorithms. In a follow-up study, the proposed model will be investigated further to identify the physiological meanings of the ECG features, such as the QS slope and RR interval, when used for personal authentication purposes.

## Figures and Tables

**Figure 1 sensors-23-01230-f001:**
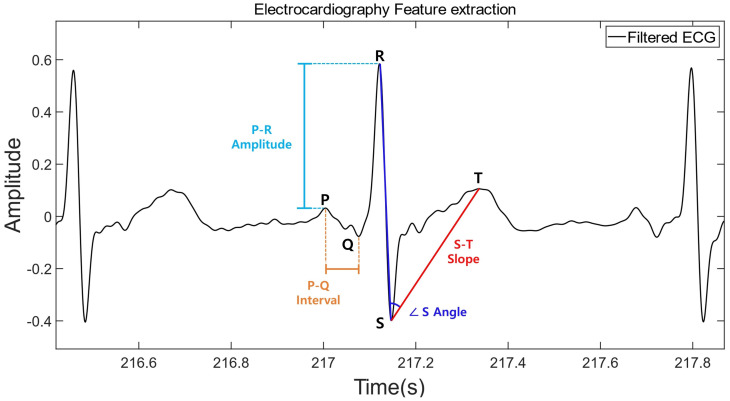
ECG feature extraction. Features were extracted from the amplitudes, intervals, angles, and slopes of the P, Q, R, S, and T peaks and the combinations of their peak points.

**Figure 2 sensors-23-01230-f002:**
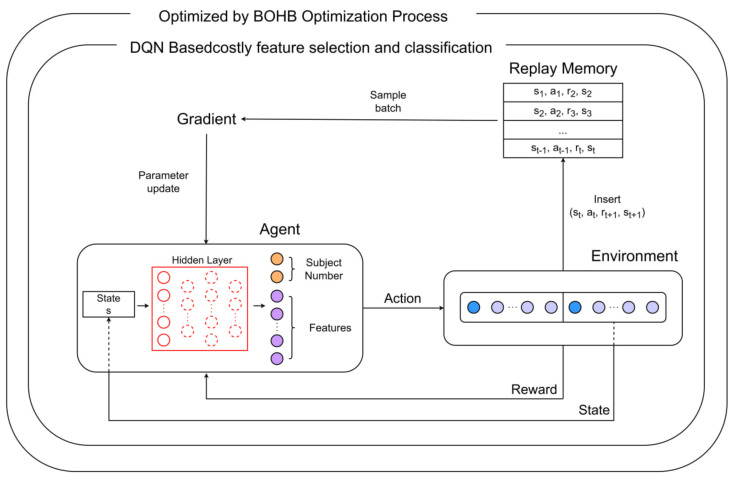
Costly feature selection and classification model based on the reinforcement learning algorithm. $Q_{class}$ denotes the optional action with which the feature selection and classification are performed. Without $Q_{class}$, only the feature selection task is conducted.

**Figure 3 sensors-23-01230-f003:**
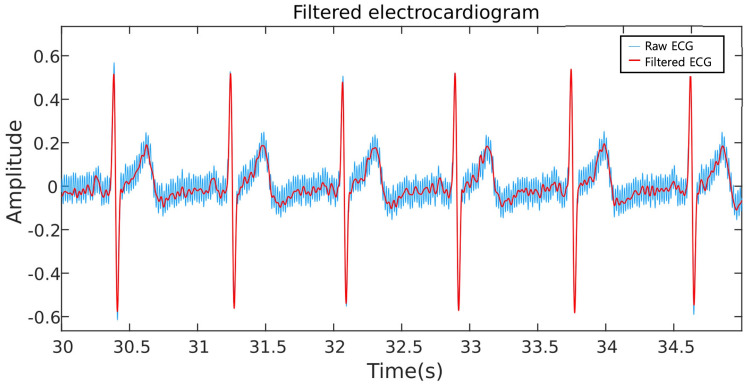
Noise–canceled electrocardiogram (ECG) signal using an FIR filter. Note that the noisy components in the raw ECG signal (in blue color) are attenuated to derive the actual signal (in red color).

**Figure 4 sensors-23-01230-f004:**
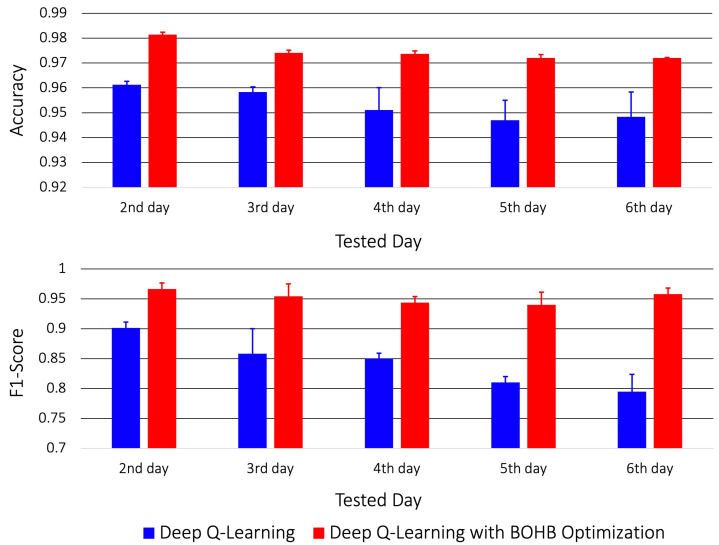
Testing results of RL based on the costly feature selection algorithm calculated using the accuracy (**upper**) and F1-score (**lower**) and plotted as a function of tested days.

**Figure 5 sensors-23-01230-f005:**
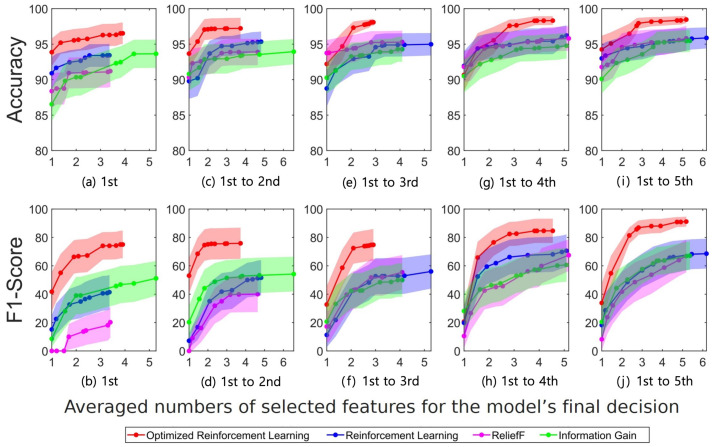
Support vector machine (SVM) results using the costly features. The accuracy and F1-score values across all subjects are illustrated with their variances in shades (corresponding to the training days). Each training dataset includes the electrocardiogram (ECG) signals recorded on the (**a**,**b**) 1st day, (**c**,**d**) 1st–2nd days, (**e**,**f**) 1st–3rd days, (**g**,**h**) 1st–4th days, and (**i**,**j**) 1st–5th days.

**Figure 6 sensors-23-01230-f006:**
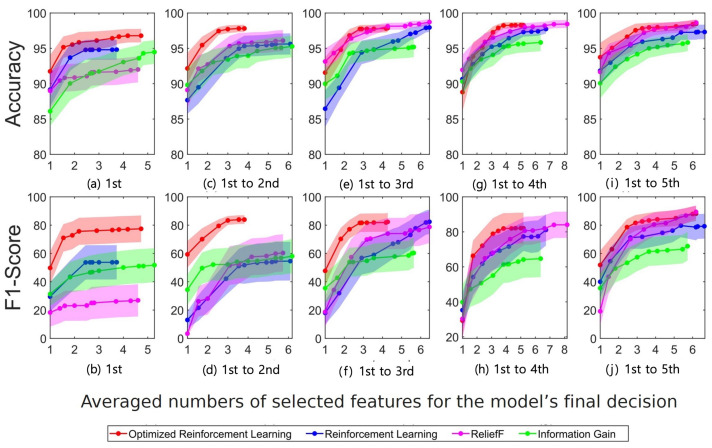
Random forest (RF) results using costly features. The accuracy and F1-score values across all subjects are illustrated with their variances in shades corresponding to the training days. Each training dataset includes the ECG signals recorded on the (**a**,**b**) 1st days, (**c**,**d**) 1st–2nd days, (**e**,**f**) 1st–3rd days, (**g**,**h**) 1st–4th days, and (**i**,**j**) 1st–5th days.

**Figure 7 sensors-23-01230-f007:**
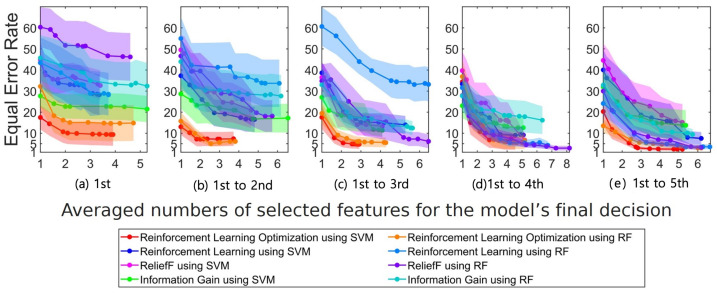
Equal error rate (EER) results using the costly features with SVM or RF. The EER values across all subjects are illustrated with their variances in shades (corresponding to the training days). Each training dataset includes the ECG signals recorded on the (**a**) 1st day, (**b**) 1st–2nd days, (**c**) 1st–3rd days, (**d**) 1st–4th days, and (**e**) 1st–5th days.

**Figure 8 sensors-23-01230-f008:**
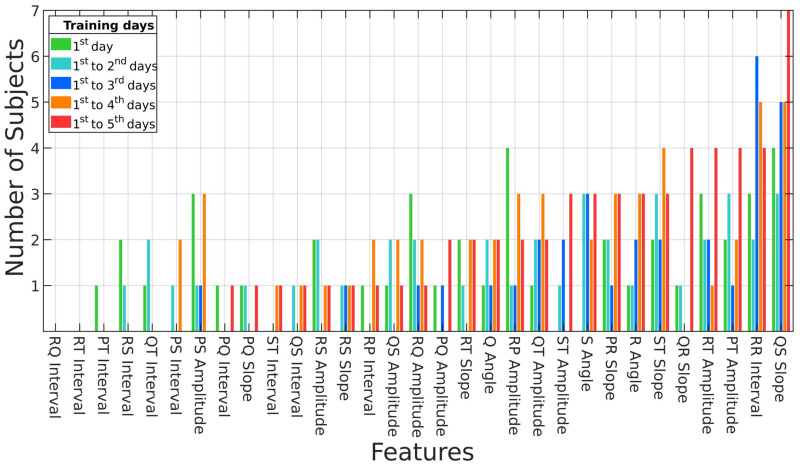
Feature selection results based on optimized RL.

**Table 1 sensors-23-01230-t001:** Features extracted from an electrocardiogram (ECG) signal.

	Features
Amplitude	R–P Amplitude R–S Amplitude R–T Amplitude P–S Amplitude P–T Amplitude S–T Amplitude R–Q Amplitude Q–T Amplitude Q–S Amplitude P–Q Amplitude
Interval	R–P Interval R–Q Interval R–S Interval R–T Interval P–Q Interval P–S Interval P–T Interval Q–S Interval Q–T Interval S–T Interval R–R Interval R–T Interval
Slope	P–R Slope R–S Slope S–T Slope Q–R Slope P–Q Slope Q–S Slope
Angle	Q Angle R Angle S Angle

**Table 2 sensors-23-01230-t002:** Search space of the DQN hyperparameters to be optimized using the BOHB algorithm.

	Range	Min	Max	Default
Hyperparameter				
Number of layers		1	4	2
Numbers of nodes in each layer		16	64	32
Learning rate		0.001	0.1	0.01
Optimizer		Adam, SGD, RMSprop		
SGD momentum		0	0.99	0.9

**Table 3 sensors-23-01230-t003:** The number of beat data for each subject from Day 1 to Day 6.

Subject No.	Day 1	Day 2	Day 3	Day 4	Day 5	Day 6
1	3055	3037	3067	3071	3132	2931
2	2839	3313	3294	3231	3181	3075
3	2923	3150	2949	3606	2962	2805
4	3130	2925	3339	3075	2832	3099
5	3021	3423	3129	2982	3034	3399
6	3622	3399	3131	2931	3178	3147
7	3093	2955	3172	3062	3264	3414
8	3303	3321	3284	2751	2931	3255
9	2667	2771	3034	2806	2994	3063
10	3007	3377	2898	3131	2985	3327
11	3429	3693	3281	3522	3367	3491
Total	34,089	35,364	34,578	34,168	33,860	35,006

## Data Availability

The data are not publicly available due to ethical issues.

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
