# Peer review of "Intelligent Feature Selection for ECG-Based Personal Authentication Using Deep Reinforcement Learning"

_sensors, 2023, doi:10.3390/s23031230_

Round 1
Reviewer 1 Report
In the manuscript, the authors implemented a reinforcement learning algorithm on ECG signals for the personal authentication application. The topic is overall interesting, and the manuscript is well organized. However, the dataset is limited and lacks enough evidence to support the application of personal authentication using ECG signals. I would suggest a minor revision before the manuscript can be considered for publication. The detailed questions are listed below:
1. As the authors mention in the manuscript, ECG signals are affected by various factors such as age, gender, physical conditions, and so on. Although this provides a higher level of security, signal reproducibility could be an issue. Can the authors provide evidence that a specific person can be identified under various conditions? For example, the person can still be identified after a heavy workout. And how well this algorithm is compared with other ones?
2. Can the authors provide more information such as age, gender, and weight regarding the 11 subjects? People of different ages and genders will give ECG signals with a more significant difference. In other words, people with similar conditions could give similar ECG signals. Without this information, it is very hard to tell whether using ECG signals for personal authentication is practical. Can the authors provide evidence showing that this method works well to identify subjects with similar conditions? And how well this algorithm is compared with other ones regarding this issue?
3. The authors should pay more attention to the figures. For example, the x-axis title is missing in both Figure 6 and Figure 7.
Author Response
Please reference the attached file.

Reviewer 2 Report
Authors used DRL to select the optimal parameters from ECG to perform the personal authentication. There are some problems in this article that have to be revised or clarified.
1. In Abstract, “Experiment results…for a wearable device intended to be used as an authentication system”. But, I do not find any specifics of DRL (Proposed method) to show that this system could be embedded in a wearable device, like RAM, power consumption, run time et al.
2. Just I know that authors used BOHB to optimize the structure of RL (DQN), RL to select cost features, ML to authenticate person. If yes, what’s Fig.4 meaning? Do the target cost features have be defined? Do the different ML models use the different cost features or the same cost features? Authors are how to get the validation results of RL in Fig. 4.
3. In Fig. 4, the results show that the more data, the lower accuracy in training. This is a common sense. The question is what’s the results of testing?
4. In Fig. 5 and 6, what’s the unit in x-axis? Are they the testing results? Authors should describe them more clear.
5. Authors do not show the number of training, validation and testing.
6. The final cost features should be described.
7. In Fig. 8, the optimal cost features depend on different training days and subjects. I think that these results are hard to be implemented in the work of personal authentication. When the more persons have to be authenticated with the more samples (more days), the cost features will be the differences.
8. Authors should perform the bench mark with open ECG datasets, like as MIT-BIH.
Author Response
Please reference the attached file.

Reviewer 3 Report
This is a good technical research paper. The objective is coherent with the proposed method and experimental results. However, several improvement are suggested as the following:
1. Refer to the Introduction section (Paragraph 5): Among the algorithms used to search costly features, .....
Please review (research line) those algorithms in term of their strengths and limitations preferably in comparison table.
Are the ReliefF and IG categorized as conventional or costly feature extraction? Please clarify.
2. Discuss the motivation why Reinforcement Learning (RL) algorithm is used in this study and not others (as reviewed in no.1 above).
3. Refer Figure 1.
Please elaborate Figure 1 so that it will reflect the methodology described in subsection 2,2 - 2.7.
4. It is recommended to have pseudocode algorithms to summarize all processes discussed in subsection 2.2 to 2.7.
5. Add some explanations on the reliefF and Information Gain in the literature review section.
6. It is recommended to have more results on several optimization models to be evaluated together with the BOHB model).
7. Please add result on the Vanila RC model (stated in the abstract) in Fig. 6 and Fig. 7.
Author Response
Please reference the attached file.

Reviewer 4 Report
see attachment

Author Response
Please reference the attached file.

Round 2
Reviewer 2 Report
Authors used DRL to select the optimal parameters from ECG to perform the personal authentication. There are some problems in this article that have to be revised or clarified.
1. For “comment 1”, authors do not reply to this comment. Please, according to my comment, authors should describe the hardware specific of proposed method more clear to support that it could be developed as a wearable device.
2. For “comment 3”, My suggestion is to delete this results because the performances of model depend on the testing results, not training results. Thus, my comment is to show the testing results in Fig. 4.
3. For “comment 5”, authors do not reply to my comment. My mean is to show the numbers of training and testing samples.
Author Response
Thank you for the comment. Please refer to the attached file.